# Integrated Phenotypic and Transcriptomic Analyses Unveil the Antibacterial Mechanism of Punicalagin Against Methicillin-Resistant *Staphylococcus aureus* (MRSA)

**DOI:** 10.3390/foods14213589

**Published:** 2025-10-22

**Authors:** Yiming Wang, Tianyu Yin, Mengyan Qian, Balarabe B. Ismail, Zhipeng Zou, Xinhui Zhang, Qiao He, Mingming Guo

**Affiliations:** 1Institute of Food Processing Engineering, College of Biosystems Engineering and Food Science, Zhejiang University, Hangzhou 310058, China; 22313078@zju.edu.cn (Y.W.); 22313079@zju.edu.cn (T.Y.); qmy1211@126.com (M.Q.); bbismail@zju.edu.cn (B.B.I.); zouzp0601@163.com (Z.Z.); 12213076@zju.edu.cn (X.Z.); heqiao@zju.edu.cn (Q.H.); 2Future Foods Laboratory, Innovation Centre of Yangtze River Delta, Zhejiang University, Jiashan 414100, China; 3Fuli Institute of Food Science, Zhejiang University, Hangzhou 310058, China

**Keywords:** food safety, natural antimicrobial, cell envelope disruption, fatty acid metabolism, amino acid metabolism

## Abstract

The growing emergence of multidrug-resistant bacterial pathogens drives the need for new antibacterial agents. Punicalagin exhibits efficacy against methicillin-resistant *Staphylococcus aureus* (MRSA), but its specific antibacterial mechanisms remain unclear. This study unveiled the specific antibacterial mechanism of punicalagin against MRSA via phenotypic and transcriptomic analyses. Punicalagin was found to induce severe cell wall damage and membrane disruption. Competitive binding assays identified lipoteichoic acid (LTA) as a potential target, and transcriptomic analysis further revealed that punicalagin downregulated key genes involved in cell wall synthesis (*mur*A, *mur*E) and LTA biosynthesis (*dlt*A-D), consistent with the disruption of the cell wall. Additionally, punicalagin disrupted membrane homeostasis by inhibiting fatty acid synthesis (*fab*D, *fab*Z) and amino acid metabolism (*dap*A, *dap*B), leading to increased membrane permeability, which aligned with the phenotypic manifestations of membrane damage. Collectively, this work links phenotypic changes to specific gene expression patterns, unveiling that punicalagin inactivates MRSA via the multi-pathway regulation of the cell wall (LTA) and membrane function—providing insights for combating antibiotic-resistant pathogens in food safety and clinical settings.

## 1. Introduction

Foodborne diseases remain a persistent global public health problem. *Staphylococcus aureus* (*S. aureus*) is a major pathogen among the most common causes of foodborne illnesses worldwide, frequently resulting in mild to severe food poisoning and acute gastroenteritis. In China, 20–25% of foodborne bacterial outbreaks are associated with *S. aureus* contamination [1]. Of greater concern is the rise in methicillin-resistant *S. aureus* (MRSA), which exhibits broad-spectrum resistance to antibiotics. This resistance is primarily mediated by the *mec*A gene that encodes penicillin-binding protein 2a (PBP2a)—a protein with low affinity for β-lactam antibiotics [2]. MRSA also enhances its survival through biofilm formation and metabolic adaptation to stress, making it resistant to most conventional antibiotics and a critical threat to food safety and clinical treatment. Wang et al. revealed that 10.9% of ready-to-eat food-derived *S. aureus* in China are MRSA, with 60% of these MRSA strains harboring multiple virulence genes and 26.4% showing multidrug resistance [3]. These findings indicate MRSA not only can cause traditional food poisoning (e.g., vomiting, diarrhea), but their resistance to first-line antibiotics (e.g., β-lactams) may complicate clinical treatment for vulnerable populations, such as the elderly or immunocompromised individuals. In addition to its clinical implications, MRSA poses severe challenges to food manufacturing and supply chains, posing significant risks to the food industry through cross-contamination, product recalls, and reduced consumer confidence [4]. The urgent need to combat MRSA stems from its dual threats to public health and food industry sustainability, and this need has driven the search for novel antibacterial agents with distinct mechanisms from traditional antibiotics.

In response to increasing consumer awareness of food safety and concerns about chemical preservatives, natural antimicrobials, especially plant-derived antimicrobial compounds, have emerged as promising alternatives [5]. Pomegranate (*Punica granatum* L., *Punicaceae*) is renowned for its nutritional and medicinal properties, with various parts (fruit, seeds, and peels) used in traditional medicine and functional foods [6]. Punicalagin, a major ellagitannin isolated from pomegranate peels, has demonstrated multiple bioactivities, including antibacterial, anti-inflammatory, and antioxidant effects [7]. It has been reported to exhibit antimicrobial activity against several pathogenic bacteria, including *S. aureus*, *Salmonella*, and *Vibrio parahaemolyticus* [8,9,10]. However, existing research on punicalagin’s activity against MRSA remains limited. Mun et al. demonstrated that punicalagin enhances oxacillin efficacy against MRSA by downregulating *mec*A (MIC = 0.8 mg/mL) [11]. Yet this study only focused on phenotypic synergy and lacked mechanistic insights into how punicalagin affects MRSA’s cell envelope (e.g., peptidoglycan/LTA structure); Wu et al. identified punicalagin as a SrtA inhibitor that reduces MRSA virulence (e.g., adhesion, biofilm), but its bactericidal mechanism and impact on MRSA’s cell wall/membrane metabolism remain unknown [12].

These studies collectively highlight critical gaps in our understanding of punicalagin’s action against MRSA: (1) Existing studies focus on morphological observations, lacking insights into key pathways (e.g., cell wall synthesis, metabolic regulation) that drive its antibacterial effect; (2) To our knowledge, no studies have linked phenotypic disruptions (e.g., membrane damage, cell wall deformation) to specific gene expression changes in MRSA, hindering the understanding of how punicalagin targets MRSA-specific resistance mechanisms; (3) Moreover, transcriptomic-level evidence to systematically elucidate punicalagin’s antibacterial mechanism against MRSA is still lacking—a gap that limits our ability to translate this compound into practical applications.

To address these gaps, this study used a combination of phenotypic analysis and transcriptomic profiling to elucidate the antibacterial mechanisms of punicalagin against MRSA. By capturing dynamic changes in structural alterations, functional disruptions, and gene expression in bacterial cells, we aimed to provide mechanistic insights linking phenotypic damage to specific pathways—including those related to cell wall integrity, membrane homeostasis, and key biosynthetic processes. This multi-level approach not only clarified the mechanism of punicalagin but also established punicalagin as a promising candidate for combating MRSA, offering new perspectives for the development of natural antibacterial agents.

## 2. Materials and Methods

### 2.1. Materials and Reagents

Punicalagin was purchased from Shanghai Yuanye Biotechnology Co. Ltd. (Shanghai, China). Methicillin-Sensitive *Staphylococcus aureus* (MSSA) (ATCC 25923) and MRSA (ATCC 43300) were purchased from the Guangdong Microbial Strain Collection Center (Guangzhou, China). Nutrient broth (NB), plate count agar (PCA), and Eosin methylene blue (EMB) agar were purchased from Qingdao Haibo Biotechnology Co. Ltd. (Qingdao, China). Baird-Parker (BP) agar and Nile Red were purchased from Shanghai Dibao Biotechnology Co. Ltd. (Shanghai, China). Rhodamine 123 was purchased from Shanghai Aladdin Biochemical Technology Co. Ltd. (Shanghai, China). An alkaline phosphatase (AKP/ALP) kit was purchased from Hangzhou Fei Shi Er Biotechnology Co. Ltd. (Hangzhou, China). The 2′,7′-dichlorofluorescin diacetate (DCFH-DA) fluorescent probe was purchased from Beijing Solebo Technology Co. Ltd. (Beijing, China).

### 2.2. Preparation of Bacterial Suspensions

MSSA and MRSA were used in this study. Single colonies of the two strains were obtained by streaking on BP agar and incubated at 37 °C for 18 h. The single colonies were then inoculated into NB medium and incubated at 37 °C with shaking at 150 rpm for 18 h until the stationary phase was reached. After incubation, cells were collected by centrifugation at 8000× *g* for 10 min, and the cell precipitates were washed twice with 0.85% sterile physiological saline. After adjusting the bacterial suspension to an optical density at 600 nm (OD_600_) of 0.5 using 0.85% sterile physiological saline, the bacterial concentration was verified to be approximately 10^6^ CFU/mL via the plate colony counting method, ensuring consistent inoculum size across experiments.

### 2.3. Antibacterial Effects of Punicalagin Against MSSA and MRSA

#### 2.3.1. Determination of Minimum Inhibitory Concentration and Minimum Bactericidal Concentration

The minimum inhibitory concentrations (MICs) of punicalagin were evaluated based on the 2-fold dilution method [13]. Briefly, logarithmic phase cultures of all the bacterial strains were diluted to 10^6^ CFU/mL with fresh NB medium. Then, punicalagin was 2-fold serially diluted in NB medium to obtain an appropriate concentration range in a 96-well plate. Next, the obtained solution was mixed with the diluted bacterial suspension at a ratio of 1:1 and incubated at 37 °C for 24 h. The absorbance of the bacterial suspension at 600 nm was measured using a microplate reader. The MIC was defined as the lowest concentration of antibacterial agent at which the OD_600_ nm was below 0.1. Thereafter, bacterial suspensions with a punicalagin concentration equal to or greater than the MIC were spread on PCA plates, followed by incubation at 37 °C for 24 h. The minimum bactericidal concentration (MBC) was defined as the lowest concentration of antibacterial agent at which no colonies were visible on the plate after incubation at 37 °C for 24 h.

#### 2.3.2. Time-Kill Curves

The time-kill curves assay was performed by the plate colony counting method to evaluate the antibacterial activity of punicalagin [14]. Briefly, MSSA and MRSA cell suspensions were diluted to 10^6^ CFU/mL with 0.85% sterile saline. Then, punicalagin was added to the bacterial suspensions to achieve final concentrations of 1×, 2×, 4×, and 8 × MIC, followed by incubation at 37 °C for 24 h. At different time points (0, 1, 2, 4, 6, 8, 12, 24 h), the samples were collected, diluted, and spread on PCA plates. After incubation at 37 °C for 24 h, colonies were counted to calculate the viable bacterial count at each time point.

### 2.4. Phenotypic Changes in MSSA and MRSA Induced by Punicalagin

#### 2.4.1. Scanning Electron Microscopy

Scanning electron microscopy (SEM) was performed to reveal the morphological alterations of MSSA and MRSA cells subjected to 8 × MIC punicalagin [15]. After being incubated at 37 °C for 1, 4, and 8 h and centrifuged at 8000× *g* for 10 min, approximately 5 mg of precipitated cells were collected and immersed in a 2.5% (*v/v*) glutaraldehyde fixative at 4 °C overnight. The samples were then post-fixed with 1% (*v/v*) osmium tetraoxide for 1 h, followed by three additional washes with 0.1 M phosphate-buffer saline (15 min per wash). Subsequently, the samples were stepwise dehydrated with ethanol (30, 50, 70, 90, 95, and 100%, *v/v*) for 15 min per gradient. A control group (untreated with ethanol, only PBS-washed) was set to verify that dehydration did not cause additional cell damage. Finally, the samples were dried in a critical point drying system, coated with gold, and visualized using a field emission scanning electron microscope (SU-8010, Hitachi Ltd., Tokyo, Japan).

#### 2.4.2. Transmission Electron Microscopy

The ultrastructure changes in MSSA and MRSA induced by punicalagin were revealed via transmission electron microscopy (TEM) [16]. Following the same fixation process as that in SEM analysis, the samples were then stepwise dehydrated with 30, 50, 70, and 80% ethanol as well as 90, 95, and 100% acetone. Approximately 2 mL of bacterial suspension (≈10^8^ CFU/mL) was used for each sample, and untreated bacterial cells served as dehydration and staining controls to ensure that structural alterations were solely induced by punicalagin treatment. Finally, the dehydrated samples were embedded in epoxy resin and sectioned into 90 nm ultrathin sections with an ultramicrotome (EM UC7, Leica, Vienna, Austria). Subsequently, the samples were stained with uranyl acetate and lead citrate before being visualized using a transmission electron microscope (JEM-F200, Jeol, Tokyo, Japan).

#### 2.4.3. Determination of Extracellular Alkaline Phosphatase

The leakage of alkaline phosphatase (AKP) was measured to determine the cell wall integrit. After MSSA and MRSA were treated with different concentrations of punicalagin (1 × MIC, 4 × MIC, and 8 × MIC) for 1 h, the supernatants of the two bacterial cells were collected by centrifugation at 8000× *g* for 10 min, and the extracellular AKP content was determined using an AKP kit.

#### 2.4.4. Determination of Extracellular K^+^

The concentration of extracellular K^+^ reveals the integrity of the cell membrane [17]. After being treated with different concentrations of punicalagin (1 × MIC, 4 × MIC, and 8 × MIC) for 1 h, the supernatant was collected by centrifugation at 8000× *g* for 10 min. The collected supernatant was filtered through 0.22 μm PVDF membrane to remove bacterial cells. An AA240 atomic absorption spectrophotometer was used to determine the concentration of K^+^ in supernatant.

#### 2.4.5. Determination of Membrane Potential

The changes in bacterial membrane potential were determined via the Rhodamine 123 (Rh 123) fluorescence probe method [18]. MSSA and MRSA cells were treated with different concentrations of punicalagin (1 × MIC, 4 × MIC, and 8 × MIC) for 1 h and centrifuged at 8000× *g* for 10 min. The absorbance of the bacterial suspension at 600 nm was then adjusted to 0.5 with 0.85% sterile physiological saline. Rh 123 solution (dissolved in phosphate-buffered solution) was added to the bacterial suspensions, and the final concentration was adjusted to 2 μg/mL. After incubation in the dark for 30 min, the stained samples were washed twice with 0.85% sterile saline and transferred to a black 96-well plate. The fluorescence was determined by an M5 full-band multifunctional enzyme-labeled instrument with excitation and emission wavelengths at 480 nm and 530 nm, respectively.

#### 2.4.6. Fourier Transform Infrared Spectroscopy Analysis

Fourier transform infrared spectroscopy (FTIR) was used to determine the conformational changes in the functional groups of bacterial membrane components [19]. After treatments with different punicalagin concentrations, cell samples were collected by centrifugation and vacuum-lyophilized. The FTIR analysis was carried out by the Nicolet iN10 FT-IR spectrophotometer (Thermo Fisher Scientific, Waltham, MA, USA), with a wavelength range of 4000−400 cm^−1^. The background obtained from pure KBr disk was automatically subtracted.

#### 2.4.7. Competitive Inhibition Assay

Peptidoglycan (PGN) and lipoteichoic acid (LTA) are the main lipid components of bacterial cell wall. Phosphatidylethanolamine (PE) and phosphatidylglycerol (PG) are the main lipid components of bacterial cell membrane and the most important phospholipids in MSSA and MRSA. Based on MIC analysis, an exogenous addition assay of PGN, LTA, PE and PG was performed [20]. 100 μL of 1 × 10^6^ CFU/mL MSSA and MRSA suspension were added to a 96-well plate, followed by the addition of 8 × MIC of anacardic acid for treatment. Then, different concentrations (1–100 µg/mL) of PGN, LTA, PE, or PG were added. The potential interaction sites of punicalagin on the cell membrane and cell wall were explored by determining the fold change in MIC.

### 2.5. Transcriptome Sampling, Sequencing and Analysis

Based on the time-kill curve obtained in 2.3.2, MRSA was treated with 8 × MIC at different time intervals. Then, transcriptomics sequencing samples were divided into four groups: CK group (no punicalagin added), PS group (8 × MIC, 1 h), PM group (8 × MIC, 4 h) and PL group (8 × MIC, 8 h). All experiments were performed in triplicate to ensure result accuracy.

#### 2.5.1. Total RNA Extraction and Analysis

After punicalagin treatment at different time intervals, treated cells were centrifuged at 8000× *g* for 10 min to collect bacterial precipitates, which were rapidly frozen with liquid nitrogen. Total RNA was extracted from control and experimental group bacteria using the Trizol method, with genomic DNA removed [21]. RNA integrity and total yield were accurately detected using a bioanalyzer.

#### 2.5.2. cDNA Library Construction and Sequencing

RNA samples were purified after passing the quality inspection to obtain mRNA, which was used as a template to synthesize the first strand of cDNA. After the second strand was synthesized, the newly generated double-stranded DNA was repaired at the end. After screening, PCR amplification and purification, the cDNA libraries were obtained [22]. Sequencing was performed on an Illumina NovaSeq 6000 platform, generating 150 bp paired-end reads.

#### 2.5.3. Differential Gene Expression Analysis

Differences in gene expression between groups were analyzed by DESeq2 (1.20.0). Padj < 0.05 & |log_2_ (fold change)| > 0 was set as the threshold for significant differential expression.

#### 2.5.4. Enrichment Analysis

ClusterProfiler (3.8.1) was used to analyze the Gene Ontology (GO) enrichment and KEGG pathway annotations of the differentially expressed genes (DEGs). A padj < 0.05 was taken as the threshold of significant enrichment.

### 2.6. Statistical Analysis

All experiments were performed in triplicate and results were expressed as means ± SD. Further analysis of variance (ANOVA) was carried out using SPSS 25.0 (SPSS Inc., Chicago, IL, USA) to determine if the differences between samples were statistically significant (*p* < 0.05). Graphs were plotted using GraphPad Prism 8.0.2 and Origin 2018.

## 3. Results and Discussion

### 3.1. Antibacterial Potential of Punicalagin Against MSSA and MRSA

The susceptibility of MSSA and MRSA to punicalagin was evaluated based on MICs and MBCs, which were 0.0625 mg/mL for MSSA and 0.5 mg/mL for MRSA. Given that the identical MIC and MBC values for both strains suggest that punicalagin effects are bactericidal rather than inhibitory, which is consistent with previous report [23]. Notably, MRSA exhibits higher antibiotic resistance to punicalagin than MSSA. The higher MIC and MBC observed for MRSA may be attributed to its multidrug-resistant phenotype, primarily conferred by the *mec*A gene encoding the penicillin-binding protein PBP2a, which has a low affinity for β-lactam antibiotic [24]. In addition, MRSA often harbors other resistance determinants, forms robust biofilms, and readily adapts to diverse environmental stresses. These combined features make MRSA more difficult to eradicate than methicillin-sensitive strains, underscoring the importance of exploring punicalagin’s interaction with MRSA-specific resistance mechanisms.

To further confirm the antibacterial efficacy of punicalagin against MSSA and MRSA, a time-kill assay was performed (Figure 1). The results demonstrated that the bactericidal effect of punicalagin on both MSSA and MRSA exhibited a dose-dependent pattern, with greater potency observed within the first hour of treatment. For MSSA, the bactericidal effect of punicalagin at 1 × MIC was negligible. However, after 1 h of treatment with 4 × MIC and 8 × MIC punicalagin, the colony counts decreased by 1.09 and 2.19 log CFU/mL, respectively, compared to the initial inoculum (10^6^ CFU/mL). After 8 h of incubation, all time-kill curves plateaued, with the colony counts remaining relatively unchanged—showing reductions of 2.37 and 4.54 log CFU/mL for the 4 × MIC and 8 × MIC groups, respectively. This plateau suggests that a subpopulation of cells may have survived or entered a dormant state, potentially due to stress-induced adaptive responses or the protective effects of biofilm-like aggregates.

The bactericidal activity of punicalagin against MRSA followed a similar pattern but was slightly lower compared to that against MSSA under the same treatment conditions (Figure 1B), aligning with the higher MIC and MBC values observed for MRSA. After 8 h of treat.

The bactericidal activity of punicalagin against MRSA followed a similar pattern but was slightly lower compared to that against MSSA under the same treatment conditions (Figure 1B), aligning with the higher MIC and MBC values observed for MRSA. After 8 h of treatment, the colony counts in the 1 × MIC, 4 × MIC, and 8 × MIC groups decreased by 0.79, 2.11, and 3.91 log CFU/mL, respectively, compared to the initial inoculum (10^6^ CFU/mL). These findings reinforce punicalagin’s antibacterial potential against both sensitive and resistant strains. Notably, the reduced sensitivity of MRSA to punicalagin highlights the complexity of its resistance mechanisms and underscores the need for further mechanistic investigations.

Collectively, these results demonstrate that punicalagin exerts clear bactericidal effects on both MSSA and MRSA in a concentration-dependent manner, while the comparatively higher tolerance of MRSA suggests the involvement of specific resistance-associated pathways that warrant deeper molecular exploration in subsequent sections.

### 3.2. Punicalagin Disrupts Surface Morphology and Ultrastructure

Changes in cell membrane morphology after punicalagin treatment were observed by SEM and TEM. The SEM results (Figure 2A) showed that untreated cells had regular spherical or oval shapes with smooth and unbroken surfaces. After treatment with 8 × MIC punicalagin for 1 h, the surface of the bacteria began to wrinkle and deform, with significant membrane disruption and leakage of intracellular substances. The cellular changes were more significant with prolonged treatment. For example, there was an increase in the number of ruptured cells and their fragments, alongside a number of the cell walls and cell membranes that were severely damaged. These observations indicate that the cell wall and membrane are key targets of punicalagin.

Similarly, the bacterial ultrastructure visualized by TEM (Figure 2B) further reaffirmed the effects of punicalagin against the tested strains. For instance, unlike the untreated control, which apparently had an intact cell envelope, MSSA exhibited cell wall or cell membrane deformity 1 h after punicalagin treatment. Moreover, the cellular contents were released from the cell periphery, suggesting that punicalagin affected the cell wall and cell membrane of MSSA and caused cell deformation. Similar ultrastructural changes were observed in MSSA cells treated with the R9F2 peptide [25]. However, in agreement with the time-kill analysis results, MRSA cells were less damaged than MSSA cells due to the presence of the *mec*A gene as well as other mechanisms such as resistance gene overexpression, target modifications, biofilm formation, and metabolic adaptability under stress conditions.

Collectively, these morphological and ultrastructural observations confirm that punicalagin exerts its antibacterial action primarily by compromising the bacterial cell envelope integrity, leading to leakage of intracellular components and cell lysis. The relatively milder damage in MRSA further highlights its enhanced structural resilience, supporting the need to elucidate molecular pathways involved in its defense response in the subsequent analyses.

### 3.3. Punicalagin Impairs Cell Wall Integrity and Membrane Function

#### 3.3.1. Punicalagin Compromises Bacterial Cell Wall Integrity

The antibacterial mechanism of punicalagin against MRSA was investigated by assessing the integrity of the cell wall and the functional characteristics of the cell membrane. AKP, an enzyme located between the cell wall and the cell membrane, was used as an indicator to determine the integrity of the cell wall [26]. Increased permeability of the cell wall results in the leakage of AKP into the extracellular space, where its activity can be measured.

As shown in Figure 3A,B, the extracellular AKP content increased dose-dependently, reaching 0.5 U/L and 0.7 U/L in the 8 × MIC-treated groups of MSSA and MRSA, respectively. The cell wall primarily functions to maintain cell shape, enhance the mechanical strength of the cell, and preserve osmotic pressure. Once compromised, it can lead to bacterial death [27]. The lower AKP leakage in MRSA compared to MSSA suggests that the thicker peptidoglycan layer of MRSA provides greater resistance to disruption.

Based on the above results, it is clear that punicalagin disrupts the integrity of the cell wall, exerting a bactericidal effect. These findings highlight punicalagin’s ability to target bacterial cell walls, even in resistant strains such as MRSA.

#### 3.3.2. Punicalagin Induces Membrane Hyperpolarization and Potassium Leakage

The cell membrane potential is the voltage difference across the cell membrane caused by the unequal distribution of ions between the inside and outside of the cell [28]. As part of the proton motive force, it is involved in the generation of adenosine triphosphate (ATP), which is the cell’s energy currency, affecting cellular metabolic activity and mitochondrial function [29]. Membrane potential, as reflected by fluorescence intensity, increases during hyperpolarization and decreases during depolarization [30]. In this study, after treatment with 8 × MIC punicalagin for 1 h, the average fluorescence intensities of MSSA and MRSA increased to 5233 arbitrary units (AU) and 4895 AU, respectively, representing a 2.93-fold and 3.32-fold increase compared to the untreated groups (Figure 3C,D). Moreover, the average fluorescence intensity increased gradually with increased concentration of punicalagin, indicating hyperpolarization induced by the antibacterial treatment.

Studies have shown that membrane hyperpolarization may result from a substantial efflux of intracellular K^+^, which occurs to maintain electrochemical balance following increased membrane permeability [31]. To investigate whether punicalagin induces similar effects, the extracellular K^+^ concentration was measured after treatment with different concentrations of punicalagin. As shown in Figure 3E,F, K^+^ leakage increased significantly in a concentration-dependent manner at 1, 4, and 8 × MIC when compared with the control group (*p* < 0.05). Notably, at 8 × MIC, MSSA presented a 2.01 ± 0.03 mg/L more K^+^ leakage than MRSA, suggesting that punicalagin causes more pronounced membrane disruption in the drug-sensitive strain.

The significant increase in K^+^ efflux indicates a loss of membrane integrity, as bacterial potassium homeostasis is critical for maintaining cellular osmotic pressure, pH balance, membrane potential, and protein synthesis [32]. The observed leakage of K^+^ is consistent with the hyperpolarization results and confirms that punicalagin compromises bacterial membrane function. Together with the previous findings on AKP leakage and membrane potential changes, these results further substantiate that punicalagin disrupts membrane homeostasis, contributing to its bactericidal action.

#### 3.3.3. Punicalagin Disrupts the Molecular Structure of Membrane Components

The cell membrane of MSSA consists of phospholipids, proteins, and polysaccharides that are essential for structural integrity, and permeability regulation [33]. FTIR was used to analyze changes in these membrane components via key spectral bands.

As shown in Figure 4, MRSA exhibited a significant increase in the 2930 cm^−1^ peak (C–H stretching of fatty acid methylene groups [34]) under 4 × MIC and 8 × MIC punicalagin, indicating elevated fatty acid content—likely a compensatory response to counteract membrane disruption, which aligns with MRSA’s intrinsic resistance (e.g., stress-induced upregulation of fatty acid synthesis [35]). In contrast, MSSA showed no notable change in this peak.

Both strains had altered peaks at 1650 cm^−1^ (amide I, proteins/peptidoglycan), 1540 cm^−1^ (amide II, proteins), and 1230–1080 cm^−1^ (phospholipid P=O/C–O–P [36]), confirming punicalagin disrupts membrane lipids and proteins. At 8 × MIC, changes were more pronounced: MSSA showed increased peaks at 1450 cm^−1^ (–CH_2_ in lipoproteins [37]), 1230 cm^−1^, and 1080 cm^−1^, while MRSA had stronger elevations in 1230–1080 cm^−1^ peaks—further verifying high-concentration punicalagin exacerbates membrane component damage.

Collectively, 8 × MIC punicalagin most significantly alters membrane components, consistent with time-kill and morphological data. It inhibits bacterial growth by disrupting membrane lipid/protein conformation, leading to hyperpolarization, increased permeability, and K^+^ efflux—with MRSA’s fatty acid elevation distinguishing it from MSSA.

### 3.4. Punicalagin Targets Lipoteichoic Acid and Membrane Phospholipids

Based on the previously presented results, punicalagin may exert its antibacterial effect by targeting one or more cell membrane components. Since the specific cell membrane components targeted by punicalagin remain unknown, it is imperative to identify the specific target via the exogenous addition of various cell membrane components. The peptidoglycan layer in MRSA is typically 20–30 nm thick and serves primarily as a protective barrier as well as a scaffold for the attachment of surface proteins and extracellular matrix [37]. Teichoic acid (TA) can be divided into two forms: lipoteichoic acid (LTA) and wall teichoic acid (WTA), which are covalently linked to the cytoplasmic membrane and peptidoglycan, respectively, accounting for 60% of the total cell wall mass [38].

As shown in Figure 5A, exogenous addition of peptidoglycan (PGN) had only a minor effect on punicalagin’s antibacterial activity at low concentrations, with a slight attenuation observed as the PGN concentration increased. In contrast, LTA significantly suppressed the bactericidal activity of punicalagin against both MSSA and MRSA, increasing the MIC by up to eightfold compared to the group without exogenous LTA addition (Figure 5B). This finding suggests that LTA is a likely key target of punicalagin, providing mechanistic insight into its ability to disrupt bacterial cell wall integrity.

Similarly, the addition of phosphatidylglycerol (PG) and phosphatidylethanolamine (PE), which are the principal phospholipids in the *S. aureus* cell membrane [39], significantly reduced the antibacterial efficacy of punicalagin in a dose-dependent manner (Figure 5C,D). This supports its direct interaction with membrane lipids. Consistent with previous reports in which PG and PE antagonized the antibacterial activities of α-mangostin, isobavachalcone [40], and chickpea-derived antibacterial peptides [41], our results further demonstrated that punicalagin exerts its bactericidal action by targeting multiple membrane and wall components, including PGN, LTA, PG, and PE. Among these, LTA appears to be the primary target, emphasizing its central role in punicalagin’s antibacterial mechanism.

Taken together, these findings reveal that punicalagin disrupts bacterial viability through multi-target interactions involving both lipoteichoic acid and membrane phospholipids. This dual-targeting mode provides a plausible explanation for its potent bactericidal effect and offers a molecular basis for understanding its broad-spectrum activity against Gram-positive pathogens.

### 3.5. Punicalagin Targets Intracellular Components and Key Metabolic Pathways in MRSA

Given the distinct phenotypic differences observed between MRSA and MSSA under punicalagin treatment—such as higher MIC values and less severe membrane disruption—we selected MRSA for subsequent omics analysis to better understand its resistance mechanisms and punicalagin’s mode of action.

#### 3.5.1. The Transcriptome Sequencing Data Exhibit High Reliability

As shown in Table 1, Q20 and Q30 represent the percentages of bases with sequencing quality values greater than 20 and 30, respectively, relative to the total number of bases in the raw data. The sequencing data volume of the samples ranged from 1.0 to 1.3 Gb, with all Q20 values exceeding 96% and all Q30 values above 90%. These results indicate that the sequencing data have high reliability and are suitable for subsequent experimental analysis. Sequence alignment results revealed that the total mapping rate of each sample was higher than 96.34%, while the multiple mapping rate was lower than 1.57% for all samples. This suggests that the reference genes were appropriately selected and that there was no contamination in the relevant experiments.

Generally, the better the parallelism of biological replicates, the more reliable the analytical results. Therefore, it is particularly important to first perform a parallelism test of biological replicates when conducting differential analysis between groups. Typically, the similarity between samples can be evaluated using sample correlation analysis and principal component analysis (PCA) results [42]. The Pearson correlation coefficient (R2) is used to measure the correlation between samples; samples with an R^2^ value closer to 1 indicate a higher degree of similarity. As shown in Figure 6A, the R^2^ values of replicate samples were all equal to 1, demonstrating an extremely high correlation and good biological reproducibility, which ensures the acquisition of valid results. In Figure 6B, PCA was performed on the gene expression values of all samples. It can be observed that there were significant differences between groups, while the within-group sample reproducibility was good—these findings confirm the reliability of the data.

Then, We analyzed the overlap of DEGs among the three biological replicates. In the PL vs. CK, a total of 1877 significant DEGs were identified, and 1596 of them (accounting for 85.3%) were commonly detected in all three replicates. This high overlap rate indicates that the DEGs induced by punicalagin are stably expressed across biological repeats rather than random fluctuations, providing a robust molecular basis for subsequent KEGG pathway enrichment and antibacterial mechanism interpretation.

#### 3.5.2. GO Enrichment Analysis of DEGs Unveils Targeting of Intracellular Components and Molecular Functions in MRSA

Transcriptomic analysis revealed 1877 significantly differentially expressed genes (971 up-regulated and 906 down-regulated) in the MRSA PL group (8 × MIC, 8 h) compared to the control (CK) (Table 2). As the number of differentially expressed genes (DEGs) increased with treatment duration, the PL vs. CK group was chosen for in-depth transcriptomic investigation (Figure 7A).

Further GO annotation and categorization of the DEG in the PL vs. CK gene sets were performed to determine the main biological functions of the screened differential genes (Figure 7B).

In molecular function (MF), most DEGs were enriched in small molecule binding, specifically targeting nucleotides (e.g., ATP), nucleoside phosphates, and anions (Figure 7B,C). These molecules are critical for MRSA’s basic physiology: ATP acts as an energy donor for metabolic enzymes, while nucleoside phosphates support ribosomal assembly—suggesting punicalagin disrupts these essential molecular interactions to impair metabolism.

In cellular component (CC), DEGs were significantly concentrated (*p* < 0.001) in “intracellular”, “intracellular fraction”, and “protein-containing complex” (e.g., ribosomes for protein synthesis). This aligns with TEM-observed cytoplasmic retraction (Section 3.2), confirming punicalagin’s targeting of intracellular structures.

In biological process (BP), DEGs participated in gene expression (regulating protein production), macromolecular biosynthesis (maintaining cell structure), and oxidation-reduction processes (controlling energy metabolism). Their disruption directly explains MRSA’s reduced proliferation and structural damage. The time-kill curve assays (Section 3.1) quantitatively demonstrated the diminished bacterial growth rate, while the structural impairment was visually corroborated by SEM and TEM observations (Section 3.2), which captured evident cell wall shrinkage in treated cells.

Collectively, punicalagin exerts bactericidal effects by targeting MRSA’s intracellular components and protein complexes, with this mechanism strongly supported by SEM/TEM-visualized ultrastructural damage.

#### 3.5.3. KEGG Pathway Analysis Identifies DEG-Linked Key Metabolic Perturbations in MRSA

To further dissect the metabolic pathways and regulatory networks associated with DEGs between the PL group and CK group, KEGG pathway enrichment analysis was performed to annotate additional cellular functions of these DEGs. A total of 70 pathways were enriched in the CK vs. PL gene set; among these, a scatterplot visualizing the 20 pathways with the highest significance is presented in Figure 8. The enrichment results demonstrated that DEGs were significantly concentrated in several key pathways critical for MRSA survival, including: (1) ribosome, (2) teichoic acid biosynthesis, (3) fatty acid degradation, (4) valine, leucine and isoleucine degradation, (5) RNA degradation pathway.

Specifically, deregulation of ribosome directly results in a decrease in the synthesis of proteins, thereby affecting normal metabolic activities. Moreover, deregulation of the teichoic acid biosynthesis reduces the structural integrity of the cell wall and affects cell division and morphology [43]. Additionally, suppression of the fatty acid degradation pathway not only compromises energy homeostasis under carbon source limitation but also disrupts the functional assembly of biofilms through perturbation of membrane architectural integrity [44]. Meanwhile, deregulation of the branched-chain amino acid biosynthesis pathway indicates that 8 × MIC punicalagin may inhibit branched-chain amino acid synthetase activity, thereby prompting MRSA to compensate by enhancing this pathway to ensure a sustained supply of branched-chain amino acids [45]. Furthermore, deregulation of the RNA degradation pathway markedly impairs translational efficacy in MRSA, concomitantly elevating mistranslation frequency [46].

Taken together, transcriptomic data show high reliability (Q20 ≥ 96%, replicate R^2^ = 1, PCA group separation); GO/KEGG enrichment reveals DEGs concentrate in ribosome metabolism, LTA biosynthesis, and fatty acid degradation pathways. These results link punicalagin’s action to intracellular metabolic perturbations, laying a foundation for dissecting its multi-target mechanism.

### 3.6. Differentially Expressed Genes Analysis Reveals Multiple Targets of Punicalagin in MRSA

In order to further find and understand the related DEGs that play a role in influencing the function and metabolic pathway of MRSA after punicalagin treatment, we analyzed the differentially expressed genes and reached the following conclusions.

#### 3.6.1. Punicalagin Downregulates Ribosomal Subunit Genes to Inhibit Translation

Ribosomes are large ribonucleoprotein particles that, in all species, consist of two subunits. In bacteria, the subunits are named 30S (small subunit) and 50S (large subunit), which together form the 70S ribosome. Ribosome assembly begins with rRNA transcription and proceeds through ribosomal protein synthesis [47]. In contrast, punicalagin treatment for 8 h inhibited the largest number of genes expressed in the ribosome, with 18 and 26 differentially expressed genes downregulated in the 30S and 50S subunits, respectively. As shown in Table 3, genes encoding the 30S and 50S subunits of ribosomes, such as *rps*B, *rps*D, *rps*E, *rps*F, *rps*G were significantly down-regulated after punicalagin treatment, and the subunits regulated by these genes bind to the precursor forms of rRNA and ribosomal proteins, preventing the normal assembly sequence of ribosomes. Therefore, the inhibition of ribosomal subunit synthesis may be the target of punicalagin action, blocking its translation process, affecting protein synthesis and killing the bacteria.

#### 3.6.2. Punicalagin Suppresses Peptidoglycan and Lipoteichoic Acid Synthesis in the Cell Wall

The MRSA cell wall is primarily composed of peptidoglycan and lipoteichoic acid [48]. Peptidoglycan is formed from alternating N-acetyl muramic acid and N-acetyl glucosamine residues linked by peptides. As shown in Table 3, the genes related to peptidoglycan synthesis were all down-regulated after 8 h of punicalagin treatment, among which *mur*A, *mur*E and *mur*T were down-regulated by 1.58, 1.47 and 1.41 times, respectively. This finding points to the inhibition of peptidoglycan synthesis pathway by punicalagin. It is also possible that punicalagin inhibited the synthesis of lipid II, resulting in the inhibition of peptidoglycan synthesis, and subsequent cell death.

TA is categorized into LTA and WTA, both of which play vital roles in cell growth and protection [38]. The synthesis of phosphomuramic acid polymers requires d-alanine modification, and the lack of this modification will increase their sensitivity to cationic antibacterial peptides. In addition, d-alanine is often controlled by DltA, DltB, DltC and DltD proteins [49]. As shown in Table 3, the expression multiples of *dlt*A, *dlt*B, *dlt*C and *dlt*D were decreased by 4.98, 4.73, 4.47 and 4.09 times, respectively, under punicalagin treatment for 8 h, which reflects the effect it has on d-alanine and phosphomuramic acid synthesis. It is reported that the lack of d-alanine modification will increase the sensitivity of bacteria to nisin, defensins and other cationic antibacterial peptides. MRSA strains lacking d-alanine modification could not be colonized on polystyrene or glass, and biofilm formation was impaired [50]. Based on the above results, it can be inferred that punicalagin led to cell death by inhibiting the synthesis of peptidoglycan and phosphomuramic acid in the cell wall, which further explained its destructive effect on cell wall integrity.

#### 3.6.3. Punicalagin Disrupts Fatty Acid Synthesis and Metabolism Pathways

Fatty acids play an important role in membrane structure, homeostasis and transportation [51]. In most bacteria, fatty acid biosynthesis is catalyzed by Fatty Acid Synthase II (FASII) system enzymes [52]. Most antibacterial agents target enzymes related to the fatty acid synthesis extension pathway, including β-ketoacyl ACP synthetase (FabF, FabB and FabH), β-ketoacyl ACP reductase (FabG), β-hydroxyacyl ACP dehydratase (FabA and FabZ) and enoyl ACP reductase (FabI, FabK, FabL and FabV) [53].

In particular, FabD catalyzes the initiation of reactions and the synthesis of short-chain fatty acids, whereas FabZ and FabL catalyze the reactions in the extended cycle step, an essential feature of the FASII pathway. FabF and FabH proteins play an important role in elongation condensation and biosynthesis of branched fatty acids, respectively. FabG and FabI, play decisive roles in the first reduction step of each cycle of fatty acids biosynthesis and the elongation cycle of bacteria, respectively [54].

Following punicalagin treatment, some of the DEGs were enriched in the fatty acid synthesis pathway. As shown in Table 3, the gene expression of genes encoding by FabD, FabF and FabZ proteins were down-regulated by 0.75 times, 0.48 times and 1.60 times, respectively, after 8 h of punicalagin treatment, and the inhibition of these enzymes may lead to changes in phospholipid bilayer permeability and cause instability in the cell membrane [55], further confirming the capacity of punicalagin to destroy the integrity and permeability of the cell membrane.

In addition, the *acc*B gene was significantly down-regulated by 1.69 times. This gene, *acc*B, is involved in protein interactions with AccC, AccB, AccA, AccD, FabF, FabG, FabH, FabI, and FabZ, which are essential for fatty acid biosynthesis. Sivaranjiani et al. [56] studied the antibacterial mode of α-kaempferol against *Staphylococcus epidermidis* RP62A using comprehensive transcriptomics and protein omics methods. The authors reported that the *acc*B gene was down-regulated by 2.12 times in the gene transcription of fatty acid biosynthesis, resulting in the inhibition of the fatty acid synthesis pathway and further enhancing the bactericidal performance of α-kaempferol. Therefore, it can be concluded that part of the bactericidal effect of punicalagin against MRSA is the inhibition of the synthesis of fatty acids.

#### 3.6.4. Punicalagin Alters Amino Acid Synthesis and Metabolism to Affect Cell Homeostasis

The biosynthesis of lysine in microorganisms involves two anabolic pathways, the α-aminoadipic acid pathway and diaminopimelic acid pathway, and the first step of the synthetic pathway requires the participation of diaminopimelic acid (DAP). The diaminopimelate pathway starts with L-aspartyl phosphate produced by aspartokinase. It then forms L-dihydropyridine dicarboxylic acid via reduction and condensation, which is cyclized by dihydropyridine dicarboxylic acid synthase (*dap*A). Further reduction by dihydropyridine dicarboxylic acid reductase (*dap*B) generates L-tetrahydropyridine dicarboxylic acid, ultimately leading to the formation of meso-DAP and lysine [57]. As shown in Table 3, the expression of DAP pathway-related genes such as *dap*A, *dap*B, and *dap*D were down-regulated after 8 h of punicalagin treatment, indicating that prolonged punicalagin treatment hinders the synthesis of lysine, thus affecting the synthesis of peptidoglycan in MRSA bacterial cell wall.

Based on the above results, the synthesis and metabolic pathways of multiple amino acids in MRSA cells changed after punicalagin treatment, a mechanism distinct from conventional antibiotics (e.g., β-lactams target peptidoglycan cross-linking), which indicated that punicalagin exerts its bactericidal effect by disturbing the dynamic balance of intracellular amino acid levels.

Collectively, DEG analysis confirms punicalagin targets four core pathways: downregulating ribosomal genes (*rps*B, *rps*D) to inhibit translation, cell wall genes (*mur*A, *dlt*A-D) to impair peptidoglycan/LTA synthesis, fatty acid genes (*fab*D, *fab*Z) to disrupt membrane metabolism, and amino acid genes (*dap*A, *dap*B) to disturb lysine synthesis—collectively mediating MRSA inactivation.

## 4. Conclusions

This study integrated phenotypic and transcriptomic analyses to elucidate the multi-target antibacterial mechanism of punicalagin against MRSA. The results demonstrated that punicalagin exhibits significant antibacterial activity against MRSA. Phenotypic analysis indicated that the cell wall and membrane are key targets of punicalagin.

Further mechanistic studies showed that punicalagin exerts its antibacterial effect through multiple pathways: specifically, it disrupts cell wall integrity by competitively binding to lipoteichoic acid (LTA) and downregulating LTA biosynthesis genes (*dlt*A-D) and peptidoglycan synthesis genes (*mur*A, *mur*E, *mur*T); Concurrently, it inhibits fatty acid synthesis genes (*fab*D, *fab*Z, *acc*B) at the membrane level to disrupt membrane lipid metabolism; Moreover, it suppresses ribosomal subunit genes (*rps*B, *rps*D, *rps*G) and lysine biosynthesis genes (*dap*A, *dap*B) at the metabolic level to cripple protein translation and metabolic homeostasis.

This multi-target antibacterial mechanism—distinct from that of β-lactam antibiotics—overcomes *mec*A-mediated resistance by simultaneously dismantling cell envelope integrity, energy metabolism, and biosynthetic pathways. These findings not only elucidate the mechanistic basis of punicalagin’s efficacy but also position it as a promising template for developing novel anti-MRSA agents aimed at overcoming multidrug resistance in both food safety and clinical contexts.

For practical deployment, punicalagin shows promise as a natural preservative for food decontamination, particularly in animal-derived products like meat and dairy, to control MRSA. However, its bitter taste at high concentrations may limit sensory acceptability. Future work should focus on mitigating this off-flavor, evaluating its interactions with food components (e.g., proteins) and common additives, and assessing its stability under various processing conditions. Furthermore, in vivo studies are essential to validate its safety and efficacy in complex biological systems, paving the way for its clinical and industrial translation.

## Figures and Tables

**Figure 1 foods-14-03589-f001:**
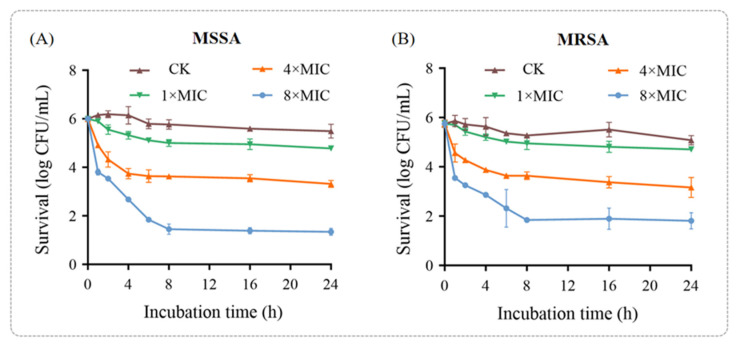
Antibacterial activity of punicalagin. The time-kill curves of (**A**) MSSA and (**B**) MRSA treated with different concentrations of punicalagin.

**Figure 2 foods-14-03589-f002:**
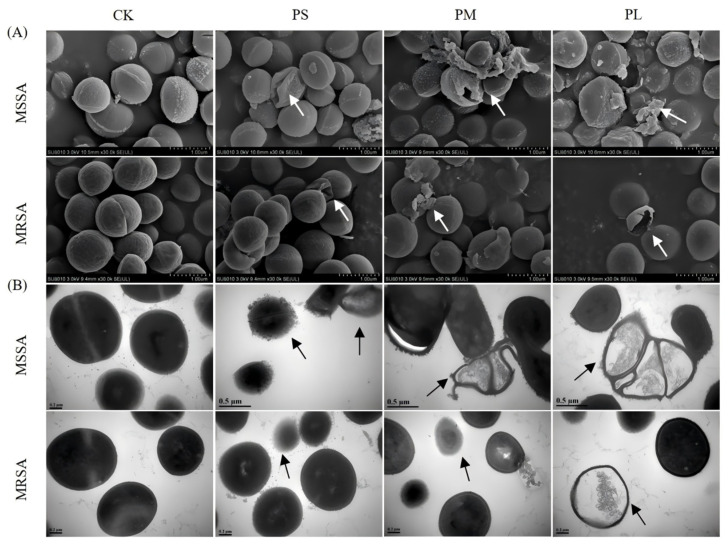
(**A**) SEM images and (**B**) TEM images of MSSA and MRSA after being treated with 8 × MIC punicalagin for different times. The arrows point to the morphological and ultrastructural changes in the cells. CK: No punicalagin, PS: Punicalagin treatment for 1 h, PM: Punicalagin treatment for 4 h, PL: Punicalagin treatment for 8 h.

**Figure 3 foods-14-03589-f003:**
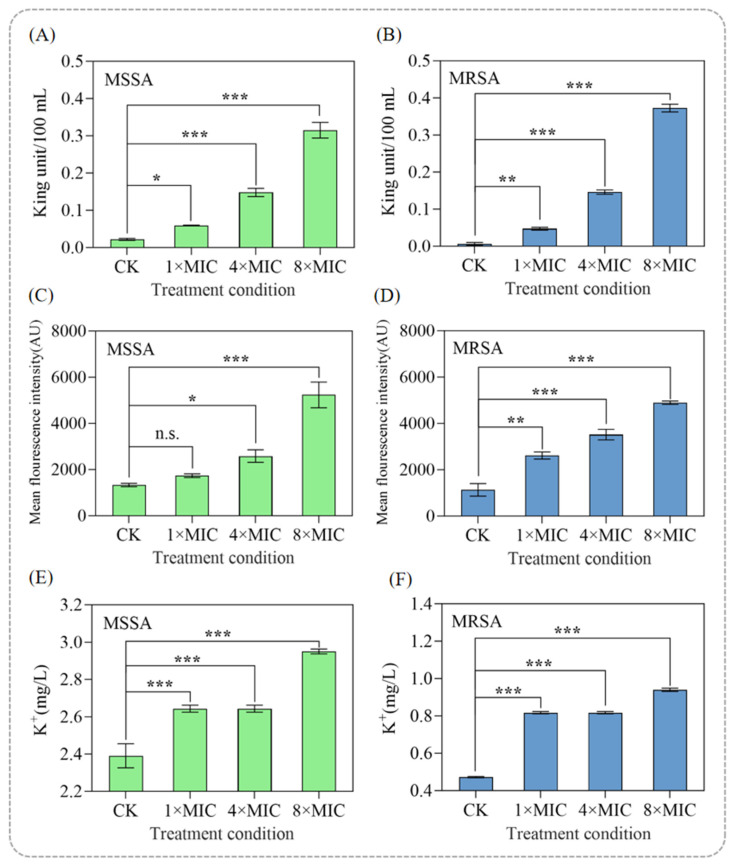
Effect of punicalagin on cell wall and membrane. AKP leakage in (**A**) MSSA and (**B**) MRSA after being treated with punicalagin for 1 h. Membrane potential of (**C**) MSSA and (**D**) MRSA after being treated with punicalagin for 1 h. K^+^ efflux in (**E**) MSSA and (**F**) MRSA after being treated with punicalagin for 1 h. All experiments were performed in triplicate, and the results were expressed as means ± SD. n.s. No significant difference * *p* < 0.05, ** *p* < 0.01, and *** *p* < 0.001.

**Figure 4 foods-14-03589-f004:**
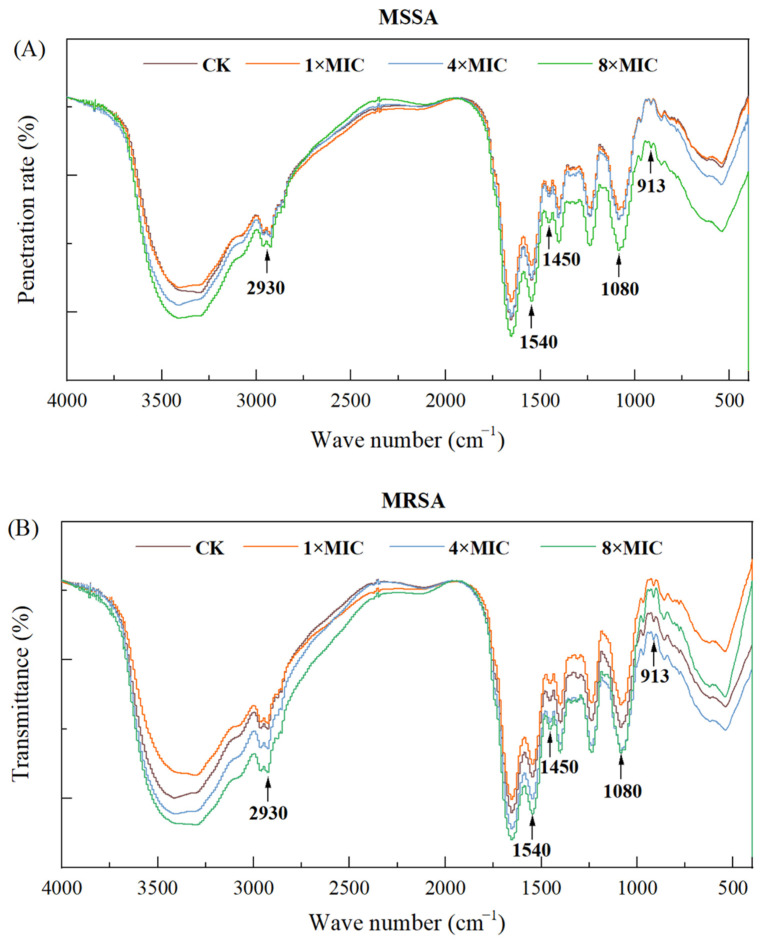
FTIR spectra of (**A**) MSSA and (**B**) MRSA upon 8 × MIC punicalagin treatment for 1 h.

**Figure 5 foods-14-03589-f005:**
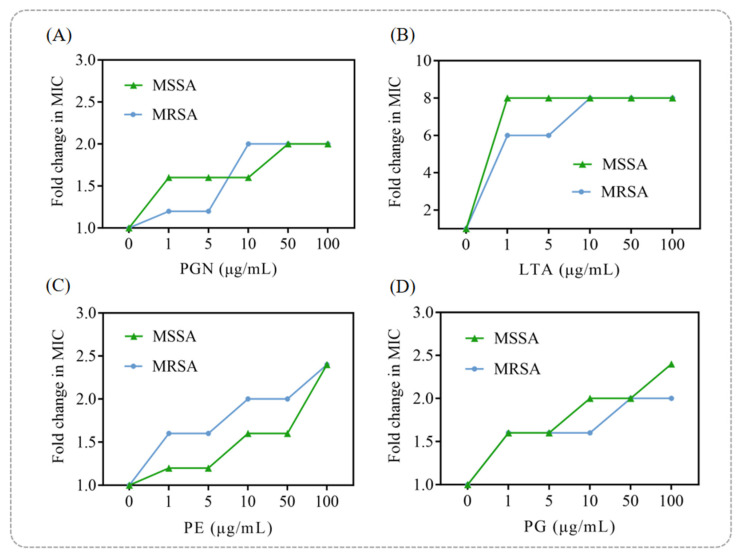
Changes in MIC of punicalagin against MSSA and MRSA cause by exogenous addition of (**A**) PGN, (**B**) LTA, (**C**) PE, and (**D**) PG.

**Figure 6 foods-14-03589-f006:**
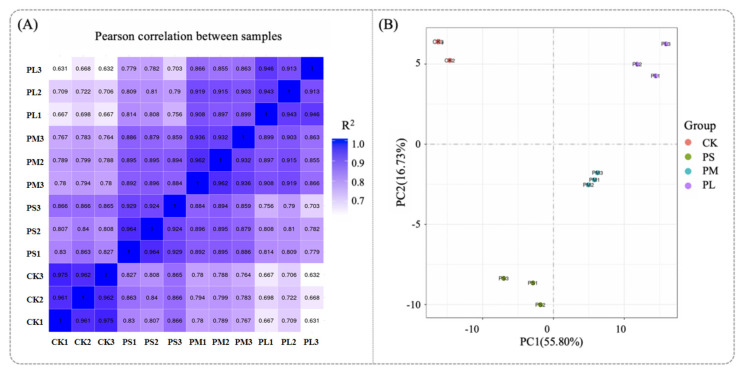
Correlation analysis (**A**) and PCA analysis (**B**); CK: No punicalagin; PS: Punicalagin treatment for 1 h; PM: Punicalagin treatment for 4 h; PL: Punicalagin treatment for 8 h.

**Figure 7 foods-14-03589-f007:**
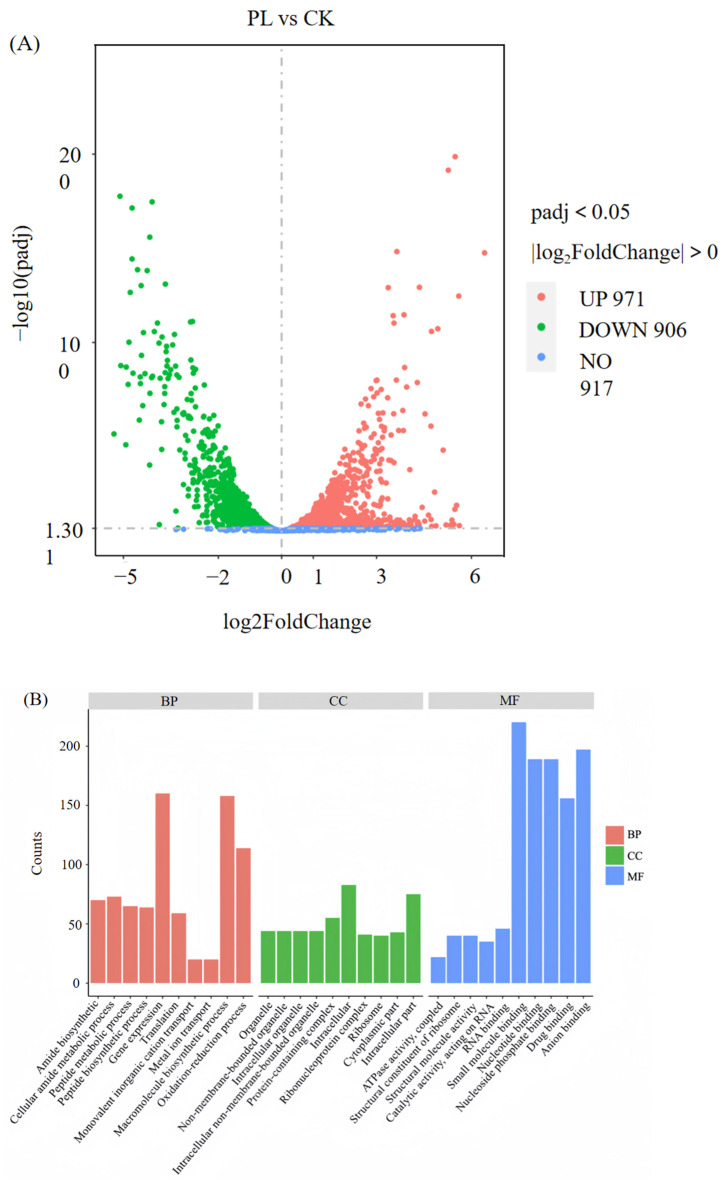
Enrichment analysis. (**A**) Volcano map showing DEGs between PL and CK. Red and green dots indicate genes whose expression significantly up-regulated or down-regulated, respectively. (*p*-adjust < 0.05). (**B**) Distribution of DEGs based on their predicted biological process, cellular component and molecular function upon punicalagin treatment. Punicalagin exerts bactericidal effects by targeting MRSA’s intracellular components and protein complexes, with this mechanism strongly supported by SEM/TEM-visualized ultrastructural damage. (**C**) GO pathway enrichment analysis of DEGs from punicalagin-treated MRSA cells.

**Figure 8 foods-14-03589-f008:**
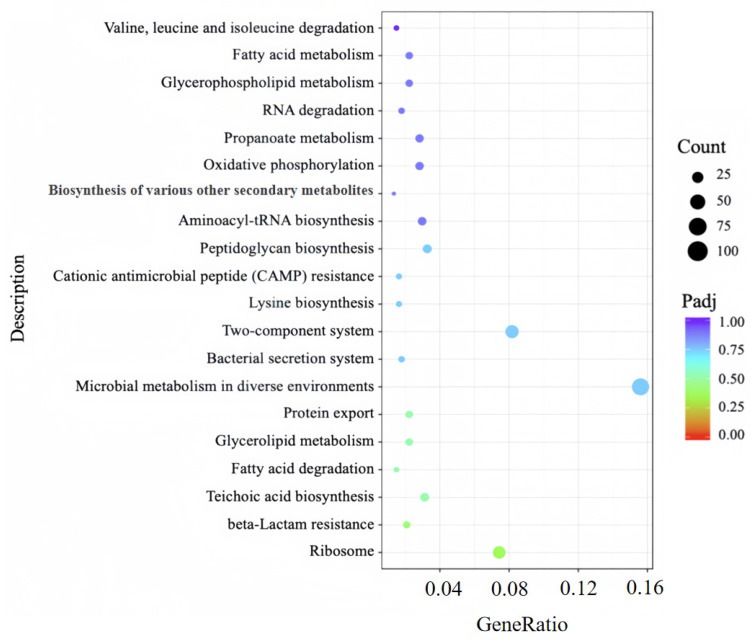
KEGG pathway enrichment analysis of DEGs from punicalagin-treated MRSA cells. CK: No punicalagin, PL: 8 × MIC punicalagin treatment for 8 h.

**Table 1 foods-14-03589-t001:** Statistical table of sample sequencing quality and sequence alignment.

Sample Names	Raw Reads (Mb)	Clean Reads (Mb)	Clean Bases (Gb)	Q20 (%)	Q30 (%)	Total Mapping Rate (%)	Multiple Mapping Rate (%)
CK-1	7.88	7.79	1.2	96.5	90.41	98.98	1.49
CK-2	7.55	7.47	1.1	96.39	90.21	98.94	1.57
CK-3	8.02	7.93	1.2	96.46	90.35	98.91	1.55
PS-1	7.89	7.80	1.2	96.43	90.33	98.52	1.52
PS-2	7.56	7.47	1.1	96.78	91.04	98.28	1.41
PS-3	7.94	7.83	1.2	96.75	90.99	96.34	1.31
PM-1	7.90	7.74	1.2	96.69	90.89	98.48	1.28
PM-2	7.79	7.71	1.2	96.63	90.76	98.46	1.21
PM-3	6.78	6.70	1.0	96.58	90.64	98.87	1.32
PL-1	9.00	8.89	1.3	96.95	91.42	98.44	1.19
PL-2	8.95	8.84	1.3	96.68	90.87	98.46	1.22
PL-3	8.01	7.92	1.2	96.74	90.80	98.56	1.27

Raw reads: The number of reads in the original data; Clean bases: The number of bases after filtering the original data (clean bases = clean reads × 150 bp); Q20: The percentage of bases with a Phred value greater than 20 among all bases; Q30: The percentage of bases with a Phred value greater than 30 among all bases; Total mapping rate: The percentage of sequencing sequences that can be successfully located on the genome out of the total number; Multiple mapping rate: The percentage of sequencing sequences that have multiple alignment positions in the reference sequence among the total number of sequences.

**Table 2 foods-14-03589-t002:** Statistical table of sample sequencing quality and sequence alignment.

Groups	Differential Expression Statistics			Threshold Value
	Up	Down	Sum	
PS vs. CK	783	759	1542	DESeq2 padj ≤ 0.05 |log_2_FoldChange| ≥ 0.0
PM vs. CK	925	917	1842	
PL vs. CK	971	906	1877	

**Table 3 foods-14-03589-t003:** List of the major metabolic pathways and the differentially expressed genes in MRSA cells upon punicalagin treatment in comparison to untreated cells.

Functional Category and Gene ID	Gene	Fold Change (Log_2_) PL vs. CK	Description
Ribosomes metabolism			
DA471_RS00895	*rps*B	−2.87	30S ribosomal protein S2
DA471_RS00330	*rps*D	−3.68	30S ribosomal protein S4
DA471_RS04005	*rps*E	−3.24	30S ribosomal protein S5
DA471_RS08515	*rps*F	−2.21	30S ribosomal protein S6
DA471_RS12395	*rps*G	−3.39	30S ribosomal protein S7
Cell wall biosynthesis			
DA471_RS02030	*mur*A	−1.58	UDP-N-acetylglucosamine 1-carboxyvinyltransferase
DA471_RS11410	*mur*E	−1.47	UDP-N-acetylmuramoyl-L-alanyl-D-glutamate--L-lysine ligase
DA471_RS10355	*mur*T	−1.41	Lipid II isoglutaminyl synthase subunit MurT
DA471_RS02415	*dlt*A	−4.98	D-alanine-poly(phosphoribitol) ligase subunit DltA
DA471_RS02420	*dlt*B	−4.73	PG: teichoic acid D-alanyltransferase DltB
DA471_RS02425	*dlt*C	−4.47	D-alanine-poly(phosphoribitol) ligase subunit 2
DA471_RS02430	*dlt*D	−4.09	D-alanyl-lipoteichoic acid biosynthesis protein DltD
Fatty acid metabolism			
DA471_RS01035	*fab*D	−0.75	ACP S-malonyltransferase
DA471_RS06400	*fab*F	−0.48	beta-ketoacyl-ACP synthase II
DA471_RS02035	*fab*Z	−1.60	3-hydroxyacyl-ACP dehydratase FabZ
DA471_RS07405	*acc*B	−1.69	acetyl-CoA carboxylase biotin carboxyl carrier protein
Amino acids metabolism			
DA471_RS03125	*dap*A	−1.15	4-hydroxy-tetrahydrodipicolinate synthase
DA471_RS03130	*dap*B	−1.14	4-hydroxy-tetrahydrodipicolinate reductase
DA471_RS03135	*dap*D	−1.27	2,3,4,5-tetrahydropyridine-2,6-dicarboxylate N-succinyltransferase
DA471_RS13535	*his*H	3.10	imidazole glycerol phosphate synthase subunit HisH
DA471_RS13515	*his*G	2.45	ATP phosphoribosyltransferase
DA471_RS13545	*his*F	2.17	imidazole glycerol phosphate synthase subunit HisF
DA471_RS13530	*his*B	2.16	imidazoleglycerol-phosphate dehydratase HisB

## Data Availability

The original contributions presented in this study are included in the article. Further inquiries can be directed to the corresponding author.

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
