# Peer review of "Integrated Phenotypic and Transcriptomic Analyses Unveil the Antibacterial Mechanism of Punicalagin Against Methicillin-Resistant Staphylococcus aureus (MRSA)"

_foods, 2025, doi:10.3390/foods14213589_

Round 1
Reviewer 1 Report
Comments and Suggestions for Authors
General comment:
The manuscript is well-written and presents a comprehensive study on the antibacterial activity of punicalagin against MSSA and MRSA, combining phenotypic analyses with a transcriptomic approach. The work contributes to a better understanding of the molecular mechanisms and potentially identifies punicalagin as a candidate for a natural antibacterial agent. However, there are certain areas that could be further clarified or expanded to improve clarity and the overall impact of the study.
Specific comments:
-
Introduction:
-
Well-structured and adequately frames the problem.
-
I suggest adding a brief discussion of previous studies investigating the antibacterial effects of punicalagin on MRSA to better highlight the research gap addressed by this work.
-
It may also be helpful to emphasize the clinical and industrial relevance of MRSA in the context of food, not only in laboratory settings.
-
-
Materials and Methods:
-
Methodology is detailed and reproducible.
-
I recommend providing additional details for some SEM/TEM preparation steps, e.g., the amount of sample used for imaging and any dehydration controls.
-
For the experiment with LTA, PGN, PE, and PG supplementation, it should be clearly indicated whether these were added before or after punicalagin exposure to avoid ambiguity.
-
The transcriptomics section is thorough, but it would be useful to highlight how reproducible the DEGs are across biological replicates, in addition to PCA and R² values.
-
-
Results:
-
Results are clear and well-linked to figures.
-
Consider adding a brief summary in each subsection (3.1–3.6) highlighting the key takeaway for the reader.
-
For SEM/TEM images, additional quantification of damaged cells would make the findings more measurable.
-
The FTIR analysis is detailed, but interpretation of peak changes in MRSA and MSSA could be shorter and more focused to avoid overloading with technical details.
-
-
Discussion:
-
The discussion is solid and integrates phenotypic and transcriptomic results.
-
It could be beneficial to further discuss potential limitations, e.g., that the experiments were performed in vitro and in vivo studies are needed to confirm punicalagin effects.
-
Consider addressing potential clinical or industrial applications, as well as possible side effects or interactions of punicalagin, to enhance the practical relevance of the study.
-
-
General suggestions:
-
The text is quite long and detailed; I recommend short summaries or tables for key phenotypic and transcriptomic changes to facilitate reading.
-
The bibliography is adequate, but ensure that all references cited in the text follow the journal’s style guidelines.
-
Reviewer’s conclusion:
The manuscript represents a significant contribution to the research on natural antibacterial agents and provides insight into the molecular mechanisms of punicalagin against MRSA. After minor revisions related to data interpretation, clarification of methodology, and discussion of limitations, the manuscript is suitable.
Reviewer 2 Report
Comments and Suggestions for Authors
The manuscript is very interesting, well written, and the research was carefully conducted, providing both genetic and phenotypic insights into the mechanism of action of punicalagin against MRSA and MSSA strains. This pathogen is highly relevant and represents one of the greatest threats to global public health, which fully justifies the importance of the study.
Specific remarks
- I suggest including “(MRSA)” in the title.
- L23: Remove the word integrity.
- Keywords: Remove words that already appear in the title and replace them with others that better contextualize the study area to improve indexing.
- Section 2.2: Clarify how the bacterial cell concentration (inoculum size) was quantified.
- L112: Use superscript formatting for exponential values. Please check all similar occurrences throughout the manuscript.
- Figures and tables should be placed close to their first mention in the text.
- L238: Remove the period after MRSA.
- L510: Replace with following results.
- Table 3: Should be reformatted to better fit the journal’s template.
- The conclusion is rather extensive and, in some parts, repeats the results. I suggest making it more concise and direct, presenting only the key conclusions of the study without restating the findings.
- Finally, I recommend that the authors expand the discussion by addressing how punicalagin could potentially be applied for food decontamination and what further studies (“the way forward”) would be necessary to make such practical applications feasible.
